# Predictors of Successful Whole-Body Hyperthermia in Cancer Patients: Target Temperature Achievement and Safety Analysis

**DOI:** 10.3390/cancers17162716

**Published:** 2025-08-21

**Authors:** Anna Lena Hohneck, Vivien Schmitz-Solheid, Deniz Gencer, Maik Schroeder, Hartmut Riess, Annette Gerhards, Iris Burkholder, Stefan Heckel-Reusser, Julia Gottfried, Ralf-Dieter Hofheinz

**Affiliations:** 1Department of Cardiology, Angiology, Haemostaseology and Medical Intensive Care, University Medical Centre Mannheim, Medical Faculty Mannheim, Heidelberg University, 68167 Heidelberg, Germany; 2European Center for AngioScience (ECAS) and German Center for Cardiovascular Research (DZHK) Partner Site Heidelberg/Mannheim, 68167 Mannheim, Germany; 3Department of Haematology and Oncology, University Medical Centre Mannheim, Medical Faculty Mannheim, Heidelberg University, 68167 Mannheim, Germany; vivien@schmitz-vulkaneifel.com (V.S.-S.); ralf-dieter.hofheinz@medma.uni-heidelberg.de (R.-D.H.); 4Klinik Oeschelbronn, Center of Integrative Oncology, Pain Management and Palliative Medicine, 75223 Oeschelbronn, Germany; d.gencer@klinik-oeschelbronn.de (D.G.); m.schroeder@klinik-oeschelbronn.de (M.S.); 5AnthroMed Öschelbronn, Centrum für Integrative Medizin, 75223 Oeschelbronn, Germany; h.riess@anthromed-oeschelbronn.de (H.R.); a.gerhards@anthromed-oeschelbronn.de (A.G.); 6Department of Health Sciences, Fulda University of Applied Sciences, 36037 Fulda, Germany; iris.burkholder@gw.hs-fulda.de; 7Heckel Medizintechnik GmbH, 73728 Esslingen, Germany; heckel@heckel-medizintechnik.de; 8Stuttgart Cancer Center (SCC)-Tumorzentrum Eva Mayr-Stihl, Zentrum für Integrative Tumormedizin, 70174 Stuttgart, Germany; j.gottfried@klinikum-stuttgart.de

**Keywords:** whole-body hyperthermia, integrative oncology, serum creatinine, safety

## Abstract

This study evaluated the effectiveness and safety of whole-body hyperthermia (WBH) in nearly 400 cancer patients treated with over 850 sessions, focusing on how often therapeutic target body temperature was achieved and what factors influenced success. WBH proved to be safe, with no serious adverse events and target temperature reached in 90% of treatments. Two factors—higher serum creatinine levels and the use of secale cornutum/galena as supportive remedy—were linked to a lower likelihood of achieving the target temperature. Common side effects included headache, mild cardiac effects, and skin reactions. These findings demonstrate that WBH is generally safe and effective as an integrative oncology treatment, but individual factors like kidney function and certain supportive medications should be considered to ensure optimal results.

## 1. Introduction

Moderate whole-body hyperthermia (WBH) has emerged as a promising integrative therapy in cancer treatment, with potential to enhance the efficacy of conventional therapies and stimulate anti-tumor immune responses [1,2,3]. Moderate WBH denotes therapeutic elevation of the body’s core temperature to therapeutic levels through heating of the entire body (not localized tumor-specific hyperthermia), typically between 38.5 °C and 40 °C, for a sustained period [4,5]. The rationale for the use of WBH in oncology is multifaceted. Preclinical studies have shown that increased temperatures can improve tumor oxygenation, enhance drug delivery, and stimulate immune responses against cancer cells [6,7,8]. Furthermore, WBH has demonstrated synergistic effects when combined with chemotherapy and radiotherapy, potentially allowing for reduced doses of these conventional treatments [9,10]. Clinical studies have explored WBH in various cancer types, including advanced or metastatic solid tumors, and chemotherapy-resistant malignancies [1,2,11,12]. For instance, a study combining WBH with gemcitabine, cisplatin, and interferon-alpha in patients with drug-resistant advanced cancers reported at least 50% tumor regression in nearly half of the cancer patients (43%) [13]. Additionally, in patients with advanced pancreatic cancer, a combination of chemotherapy with mistletoe and hyperthermia significantly improved survival in retrospective analyses [14]. Despite these promising findings, the optimal parameters for WBH administration and factors predicting treatment success remain unclear. While generally well-tolerated, WBH can cause side effects such as headache, skin reactions, and cardiovascular changes [4,15,16]. Furthermore, the extent to which the use of other integrative therapies such as mistletoe or supportive remedies could enhance the effects of WBH or mitigate side effects is also unclear [17,18]. Prospective randomized studies with oncological endpoints such as progression-free survival (PFS) and overall survival (OS) are still needed to further establish the efficacy of WBH in cancer treatment. Systematic reviews and phase II studies have described potential benefits of WBH as an adjunct treatment in advanced malignancies, but highlight significant limitations such as heterogeneity, small sample sizes, a lack of phase III data, and frequent use of combined therapy regimens [1,19,20,21]. Importantly, there is currently no consensus regarding independent predictors of efficacy. For the first time, our study provides evidence-based insights into laboratory markers (e.g., serum creatinine) and pharmacological factors that predict successful achievement of target temperature, representing a substantive contribution to the optimization of WBH protocols and patient selection.

This study aimed to assess predictive factors for successful WBH treatment—defined as achieving a target core body temperature of ≥38.5 °C—and to describe the associated adverse effects. We retrospectively included all patients who underwent moderate WBH at a single integrative oncology center over a 12-month period. Although hyperthermia is conventionally administered within a few hours before or after chemotherapy or radiotherapy to exploit synergistic effects, we prioritized safety and clarity of analysis by performing WBH independently of these intensive treatment phases. To our knowledge, this represents one of the first larger single-center analyses to examine factors linked to target temperature achievement during moderate WBH in cancer patients, together with a structured safety assessment. By focusing on a consecutive real-world cohort treated in accordance with the German WBH guideline and outside concurrent chemo- or radiotherapy phases, our study provides practical insights into WBH application and tolerability in routine clinical practice.

## 2. Materials and Methods

The present retrospective analysis includes n = 397 consecutive cancer patients, who were treated with moderate whole-body hyperthermia (855 documented treatment sessions) at the Klinik Öschelbronn between January 2018 and December 2018. No formal exclusion criteria for data collection were in place and the WBH protocols and records of all patients treated within this 12-month period were evaluated.

The primary endpoint of this retrospective analysis was the rate of patients achieving a target temperature of ≥38.5 °C. Secondary endpoints comprised the time to achieve the target temperature, safety, and the evaluation of predictive factors for achieving the target temperature.

The study was conducted according to the principles of the declaration of Helsinki and was approved by the ethical committee of the Landesärztekammer Baden-Württemberg, Stuttgart, Germany (registration number F-2020-160). Data protection was in accordance with the EU Data Protection Directive.

### 2.1. Baseline Characteristics

Demographic data including age, gender, weight, and height were collected from all study participants at baseline. Comprehensive information about each patient’s cancer diagnosis was also gathered, encompassing tumor localization and the date of initial diagnosis. Additionally, details of cancer treatment, including conventional chemotherapy and hormone therapy, as well as concurrent use of (integrative) medications, such as ongoing mistletoe therapy or supportive remedies and the use of concomitant medication of special interest (CMSI; i.e., thyroid medication, beta blockers, NSAIDs, and metamizole, which may affect the body’s thermoregulatory responses and therefore potentially hinder the attainment of target temperatures during WBH), were also recorded.

### 2.2. Whole-Body Hyperthermia

WBH was administered using moderate whole-body hyperthermia with heckel-HT2000 (Heckel Medizintechnik GmbH, Esslingen/N., Germany) with diffuse reflection-scattered IRA/B and heckel-HT3000 (Hydrosun Medizintechnik GmbH, Müllheim/Baden, Germany) with water-filtered infrared A (wIRA) according to the guideline on WBH of the German Society for Hyperthermia (DGHT) [4]. This protocol involves heating the entire body to achieve systemic elevation of core temperature, rather than localized hyperthermia focused on the tumor site. Patients were treated to achieve a target core body temperature of 38.5–39.5 °C, as measured by a rectal probe. Vital signs including heart rate, blood pressure, respiratory rate, and oxygen saturation were continuously monitored throughout the WBH procedure and for 2 h post-treatment. The heating phase lasted approximately 90–120 min, followed by a plateau phase of 60 min at the target temperature, and a cool-down phase of 60 min. WBH is generally administered twice during a one-week hospital stay and repeated based on individual decision-making. For the purpose of supporting the increase in body temperature, phosphorus D6 and secale cornutum/galena were used by the staff on an individual basis. Moreover, to support the patients in case of circulatory reactions, rosemary D3, cardiodorone, and bryophyllum, as well as gelsemium, were likewise used on an individual basis. All these supportive medications as well as the liquid infusions were recorded.

All WBH sessions were scheduled during therapy-free intervals, i.e., outside of periods with ongoing chemotherapy or radiotherapy, rather than concurrently, to avoid additive acute toxicities and confounding effects from ongoing intensive therapies and to allow side effects and temperature responses to be attributable solely to WBH.

### 2.3. Assessment of Side Effects

Cardiovascular parameters were continuously monitored throughout the WBH procedure to assess tachycardia (defined as a heart rate > 100 beats per minute (bpm)), hypotension (systolic blood pressure < 100 mmHg), and hypertension (systolic blood pressure ≥ 160 mmHg). Other side effects such as headache, fever > 40 °C, skin reactions, neurological side effects, and restlessness were systematically recorded for each patient using a standardized questionnaire. Blood samples were collected before and after WBH treatment to assess changes in serum electrolytes, liver function tests, and renal function.

### 2.4. Statistical Analysis

Quantitative parameters were summarized using descriptive statistics (N, arithmetic mean, standard deviation, median, minimum and maximum) and qualitative characteristics were evaluated in the form of frequency tables. Univariate logistic regression was performed to determine predictive factors for reaching the target temperature (38.5 °C). A generalized score test was used to test the effect of the parameter and value effects are specified as odds ratios with 95% confidence intervals. All variables showing significant associations in univariate analysis (significance level of 5%) were then included in a multivariate logistic regression model.

Generalized estimating equations (GEEs) are used for the analysis in order to take into account the correlations due to repeated measurements in a person. An autoregressive correlation structure was assumed, i.e., data points within a person that are closer together in time are more strongly correlated than data points that are further apart.

In addition, an exploratory analysis was carried out for a higher target temperature of 39 °C.

Statistical analyses were performed using the Statistical Analysis System (SAS version 9.4, SAS Institute Inc., Cary, NC, USA). Two-sided *p*-values < 0.05 were considered statistically significant.

## 3. Results

A total of 397 cancer patients underwent moderate WBH during the 12-month study period. All patients were treated as inpatients during chemotherapy-free and radiation therapy-free intervals, without receiving concurrent chemotherapy or radiation therapy during WBH sessions. The majority were female (76.6%), reflecting the predominance of breast cancer in the cohort (52.4% of all cases). The median age at cancer diagnosis was 53 years (minimum 27 years, maximum 87 years) and at the time of WBH initiation, it was 58.0 years (minimum 33 years, maximum 88 years).

More than half of the patients (54.7%) presented with metastatic disease at the start of WBH. Chemotherapy was ongoing in 10.8% of the cohort, while hormone therapy was being used by 24.9%. Integrative therapies were widespread: 85.4% received mistletoe therapy and two-thirds were on supportive remedies. Lab parameters showed normal median hemoglobin (13.1 g/dL), leukocyte counts (5.5 × 10^9^/L), and creatinine (0.8 mg/dL).

The patient population was typical of an integrative oncology center—predominantly breast cancer, high incidence of advanced disease, and widespread use of integrative therapies. The high baseline functional status (Barthel index median 100) suggests that this was a relatively fit population despite advanced cancer.

Complete baseline characteristics are shown in Table 1.

### 3.1. WBH Treatment

Across the study cohort, 855 WBH treatment sessions were performed. The primary endpoint—achieving the target core body temperature of ≥38.5 °C—was met in 770 sessions (90.1%), with failure to reach 38.5 °C occurring in only 9.9% of sessions, underscoring the technical feasibility of achieving moderate hyperthermia in most patients. Treatment characteristics were remarkably similar between successful and unsuccessful sessions, with median session duration approximately 202 min regardless of outcome. However, successful sessions achieved higher median maximum temperatures (39.1 °C vs. 38.2 °C in failures) and reached the target temperature of 38.5 °C within a median time of 94 min.

Therapy co-interventions were common throughout the treatment sessions. Mistletoe therapy during WBH occurred in 87.4% of sessions, with 35.7% of patients receiving it the evening before treatment. Concomitant medication of special interest (CMSI), including thyroid hormones, beta blockers, NSAIDs, and metamizole, was present in 77.7% of sessions. Notably, supportive remedies were frequently utilized (in 609 of 855 sessions, 71.2%), among which the most frequently used were phosphorus D6 (329, 38.5%), cardiodorone (194, 22.7%), bryophyllum (179, 20.9%), and gelsemium (156, 18.2%), followed by secale cornutum/galena (69, 8.1%) and others (141, 16.5%) (Table 2).

### 3.2. Adverse Side Effects

Overall, WBH was well-tolerated. The most frequent side effect was headache, reported in 54.9% of sessions, followed by skin reactions (11.7%), mild cardiac side effects (9.4%), and neurological symptoms (7.8%). Extreme hyperthermia (>40 °C) occurred in only 0.5% of sessions. Hemodynamic fluctuations were modest: hypotension (<100 mmHg) occurred in 15.6%, hypertension (>160 mmHg) in 6.8%. The median heart rate peak was ~110 bpm. Supportive measures were often required: saline infusion (median 1000 mL) was given in almost all cases, and oxygen therapy in 22.2% of sessions (median duration 94 min). Adverse event profiles were similar between sessions achieving and not achieving target temperature (Table 2).

Adverse events were generally mild to moderate in intensity, with most being transient headaches or minor cardiovascular changes. Given the mild overall toxicity profile, no additional exploratory analyses were performed to identify independent predictors for the occurrence of adverse events, as this was not considered a primary objective of the study.

### 3.3. Logistic Regression Analyses

To identify predictors of achieving the target temperature of ≥38.5 °C, univariate logistic regression analysis revealed three notable associations with reduced likelihood of success. Higher baseline serum creatinine emerged as a significant predictor, with each unit increase associated with substantially lower odds of reaching target temperature (OR 0.31, *p* = 0.01). Similarly, the use of phosphorus D6 was associated with reduced likelihood of success (OR 0.51, *p* = 0.01), while secale cornutum/galena use demonstrated the strongest negative association (OR 0.27, *p* < 0.001). In contrast, other patient and treatment factors—including sex, age, metastatic status, recent chemotherapy, mistletoe therapy, and CMSI—showed no significant association with temperature achievement (Table 3).

When these variables were subjected to multivariate analysis to control for potential confounding factors, only two predictors retained statistical significance as independent determinants of WBH success. Serum creatinine remained a strong predictor (OR 0.30, *p* = 0.01), while secale cornutum/galena use continued to show a robust negative association (OR 0.26, *p* < 0.001). Notably, phosphorus D6 did not maintain significance in the multivariate model, suggesting its association may have been mediated through other factors (Table 4).

### 3.4. Explorative Analysis

A post hoc analysis evaluated sessions targeting 39 °C. This higher temperature was reached in 62.0% of all sessions. Treatment times were similar to the 38.5 °C cohort (median 199 min), and adverse event rates were not significantly different, suggesting that escalation to 39 °C does not markedly increase patient risk in this setting. Again, phosphorus D6 (*p* = 0.003) and secale cornutum/galena (*p* = 0.01) were associated with lower success rates; both remained significant in multivariate analysis.

## 4. Discussion

The present study offers new, real-world insights into the feasibility, effectiveness, and safety of whole-body hyperthermia (WBH) in cancer patients. In particular, we identified independent predictors of achieving the target therapeutic temperature and evaluated the safety profile under standardized conditions. Our primary focus was placed on identifying predictors that may enhance the effectiveness of WBH.

Key findings included the following:A high success rate (90.1%) in reaching the target temperature of 38.5 °C was achieved, while a higher temperature of 39 °C was reached in 62% of patients.Serum creatinine and secale cornutum/galena emerged as independent predictors of WBH success.Concomitant medication of special interest (CMSI; i.e., thyroid medication, beta blockers, NSAIDs, and metamizole) did not significantly impact the ability to reach target temperatures.We also observed an indirect association suggesting the potential effectiveness of supportive remedies such as phosphorus D6 and secale cornutum/galena in reaching target temperatures.

There are different forms of WBH. According to the classification of the German Society for Hyperthermia, a distinction can be made between mild WBH, in which the rectally measured temperature does not exceed 38.5 °C, moderate WBH with a temperature range of up to 40.5 °C, and extreme WBH, in which this limit is exceeded [4]. Due to the extreme strain on the body, the latter should only be carried out under intensive and continuous medical supervision and is only suitable for a few, precisely defined applications in cancer therapy and chronic infections [22]. This study investigated the effectiveness and possible side effects of moderate WBH. The primary endpoint was defined as reaching a target temperature of 38.5 °C, and in an exploratory analysis, it was also analyzed at 39 °C. We observed high success rates of over 90% in reaching the target temperature of 38.5 °C, with 62% of patients achieving a higher temperature of 39 °C. Compared to a temperature of 38.5 °C, a higher temperature of 39 °C may offer potential advantages in terms of stronger therapeutic effects through more intensive stimulation of the immune system and increased blood circulation. On the other hand, a higher target temperature can be associated with stronger side effects between the two thresholds. In the present study, there was no significant difference in the occurrence of adverse side effects. Moreover, there were no serious side effects with either treatment, which corroborates the safety of WBH. To what extent 39 °C WBH could achieve better effects in terms of therapeutic outcomes compared to 38.5 °C cannot be answered on the basis of the present evaluation and warrants further investigation.

Serum creatinine and secale cornutum/galena emerged as independent predictors of WBH success. The association with secale cornutum/galena likely reflects its reactive use in cases of insufficient temperature rise, consistent with a negative association in regression modeling. The kidney is integral to thermoregulation and thermohomeostasis, primarily through its role in managing fluid and electrolyte balance, which directly impacts the body’s capacity to store or dissipate heat [23]. Creatinine levels can serve as an indicator of hydration status, potentially linking hydration to the efficacy of temperature increase during hyperthermia treatment [24,25]. Inadequate hydration may hinder the body’s ability to achieve and maintain the target temperature, as proper fluid levels are essential for efficient heat transport and regulation [26]. Elevated creatinine levels often signify compromised renal function, which can have far-reaching effects [27]. Patients with kidney dysfunction may experience reduced blood flow to various body regions, potentially impeding effective heat distribution during hyperthermia therapy. Moreover, impaired kidney function can trigger metabolic alterations that may influence the body’s heat production and regulatory mechanisms [28]. It is also important to note that high creatinine levels may not only indicate kidney injury but could also reflect other underlying health conditions, which could affect WBH responses [29]. These findings support pre-treatment hydration and renal assessment as practical measures to optimize WBH outcomes.

The use of CMSI did not significantly hinder the attainment of target temperatures. This included the use of NSAIDs or analgesics, which, due to their antipyretic properties, could theoretically interfere with reaching the desired temperature during hyperthermia treatment [30]. The same applied for oncological co-medication and supportive remedies. In particular, the use of mistletoe therapy on the previous evening did not yield substantial differences. However, an indirect association was observed suggesting possible effects of supportive remedies such as phosphorus D6 and secale cornutum/galena in reaching target temperatures. These exploratory findings require confirmation in controlled prospective trials before any clinical recommendation can be made.

While most studies have examined hyperthermia in the context of concurrent chemo- or radiotherapy, our protocol afforded a unique opportunity to assess side effects and predictors of target temperature achievement attributable specifically to WBH. This limits confounding acute interactions but may also impact comparability to studies focused on treatment synergy.

### Strengths and Limitations

This study aimed to determine predictive factors, safety, and feasibility with regard to achieving the target temperature at moderate WBH. These parameters were defined in advance and set as primary endpoints. The retrospective analysis does not allow any causal conclusions to be drawn but may inform planning of WBH, e.g., adequate hydration in patients with increased creatinine levels, etc.. Due to a lack of follow-up, no information can be provided on the impact on survival or quality of life. Furthermore, all WBH treatments were performed outside of concurrent chemo- or radiotherapy phases, a sequencing deliberately chosen to minimize overlapping acute toxicities and potential pharmacodynamic interactions, thereby ensuring that observed side effects and predictors of temperature achievement could be attributed specifically to WBH.

While the study was designed to investigate predictors of successful target temperature achievement, it did not aim to systematically identify predictors for the occurrence of adverse events. Given the overall mild toxicity profile observed, such an analysis was considered beyond the scope of the current work. Future studies involving larger cohorts and a higher incidence or severity of side effects may enable more meaningful modeling of potential toxicity predictors.

## 5. Conclusions

In this large single-center real-world analysis of 397 cancer patients undergoing a total of 855 moderate whole-body hyperthermia (WBH) sessions according to the German WBH guideline, we identified serum creatinine and the use of secale cornutum/galena as independent predictors of achieving the therapeutic target temperature of ≥38.5 °C. All treatments were performed outside of active chemotherapy and radiotherapy phases, allowing for an unconfounded assessment of safety. Adverse events were generally mild and transient, most frequently consisting of headaches, cardiovascular changes, and skin reactions.

Our findings add to the limited clinical evidence on factors influencing WBH effectiveness and may help guide patient selection and treatment planning in integrative oncology settings. In particular, the association between baseline creatinine and temperature achievement underscores the potential relevance of hydration status and renal function in optimizing WBH protocols.

While our retrospective design precludes causal inference, these results provide a basis for future prospective studies evaluating modifiable predictors of WBH success, refining patient preparation, and exploring the integration of WBH into multimodal cancer therapy.

## Figures and Tables

**Table 1 cancers-17-02716-t001:** Baseline characteristics.

	N = 397
Sex (female), %	304 (76.6)
Age at time of first diagnosis	53 (27–87)
Age at time of study inclusion	58 (33–88)
Advanced disease stage	217 (54.7)
Barthel index	100 (80–100)
Chemotherapy	43 (10.8)
Radiotherapy	161 (40.6)
Ongoing hormone therapy	99 (24.9)
Ongoing mistletoe therapy	339 (85.4)
Supportive (integrative) remedies	267 (67.3)
Tumor type, n (%)	
Breast cancer	208 (52.4)
Prostate cancer	52 (13.1)
Gynecological cancers	42 (10.6)
Colorectal cancer	30 (7.6)
Hepato-pancreatico-biliary tumors	14 (3.5)
Lung cancer	12 (3.0)
Upper gastrointestinal cancer	12 (3.0))
Urothelial cancer	10 (2.5)
Melanoma	6 (1.5)
ORL cancer	4 (1.0)
Others	7 (1.8)

ORL: otorhinolaryngological.

**Table 2 cancers-17-02716-t002:** Characteristics of whole-body hyperthermia (WBH) treatment sessions.

	N = 855 WBH Treatment Sessions	Target Temperature (38.5 °C) Achieved, n = 770 (90.1%)	Target Temperature Not Achieved, n = 85 (9.9%)
Barthel index	100 (80–100)	100 (80–100)	100 (90–100)
Chemotherapy within the last 4 weeks	70 (8.2)	62 (8.1)	8 (9.4)
Radiotherapy within the last 4 weeks	42 (4.9)	38 (4.9)	4 (4.7)
Ongoing hormone therapy	233 (27.3)	215 (27.9)	18 (21.2)
Ongoing mistletoe therapy	747 (87.4)	672 (87.3)	75 (88.2)
Mistletoe therapy on the previous evening	305 (35.7)	274 (35.6)	31 (36.5)
Median treatment duration, minutes	202 (0–321)	202 (0–321)	200 (0–245)
Maximum temperature, °C	39.1 (36.4–40.4)	39.1 (38.5–40.4)	38.2 (36.4–38.4)
Time to maximum temperature, minutes	145 (54–256)	144 (54–256)	154 (60–224)
Time to 38.5 °C	94 (9–212)	94 (9–212)	-
Fever time above 38.5 °C	124 (5–196)	124 (5–196)	-
Hemoglobin, g/dL	13.1 (7.1–16.4)	13.0 (7.1–16.4)	13.3 (10.4–15.9)
Leukocytes, 10^9^/L	5.5 (2.3–45.0)	5.5 (2.3–45.0)	5.6 (2.7–21.9)
Creatinine, mg/dL	0.8 (0.4–2.7)	0.8 (0.4–2.5)	0.8 (0.6–2.7)
Gamma-glutamyl transferase, U/L	19.0 (0.9–1881)	18.0 (0.9–1881)	20.0 (7.0–165.0)
*Concomitant medication, n (%)*	664 (77.7)	591 (76.8)	73 (85.9)
Betablocker	93 (10.9)	84 (10.9)	9 (10.6)
Thyroid hormones	153 (17.9)	130 (16.9)	23 (27.1)
NSAID	162 (18.9)	140 (18.2)	22 (25.9)
Metamizole	95 (11.1)	85 (11.0)	10 (11.8)
*Supportive* *remedies, n (%)*			
Rosemary D3	40 (4.7)	33 (4.3)	7 (8.2)
Phosphorus D6	329 (38.5)	280 (36.4)	49 (57.6)
Secale cornutum/galena	69 (8.1)	54 (7.0)	15 (17.6)
Cardiodorone	194 (22.7)	181 (23.5)	13 (15.3)
Bryophyllum	179 (20.9)	170 (22.1)	9 (10.6)
Gelsemium	156 (18.2)	143 (18.6)	13 (15.3)
Others	141 (16.5)	132 (17.1)	9 (10.6)
*Adverse side effects, n (%)*			
Cardiac side effects	80 (9.4)	74 (9.6)	6 (7.1)
Maximum heart rate, bpm	110 (49–211)	110 (49–190)	107 (66–211)
RR > 160 mmHg	58 (6.8)	47 (6.1)	11 (12.9)
RR < 100 mmHg	133 (15.6)	124 (16.1)	9 (10.6)
Headache	469 (54.9)	420 (54.5)	49 (57.6)
Fever > 40 °C	4 (0.5)	4 (0.5)	-
Skin reactions	100 (11.7)	88 (11.4)	12 (14.1)
Neurological side effects	67 (7.8)	61 (7.9)	6 (7.1)
Restlessness	52 (6.1)	48 (6.2)	4 (4.7)
Saline infusion, ml	1000 (145–2000)	1000 (160–2000)	1000 (145–1000)
Oxygen	190 (22.2)	164 (21.3)	26 (30.6)
Median duration of oxygen therapy, minutes	94 (2–99)	94 (2–99)	91 (90–98)

Data are presented as median (range) or frequencies. NSAID, Nonsteroidal Anti-Inflammatory Drug.

**Table 3 cancers-17-02716-t003:** Univariate logistic regression.

Parameter	Characteristics	*p*-Value	OR Estimator, 95% CI
Sex (N = 854)	female versus male	0.40	1.32 [0.69, 2.52]
Age at time of treatment (N = 855)	continuous	0.75	1.0 [0.97, 1.040]
Metastases (N = 855)	existant versus none	0.71	1.11 [0.63, 1.97]
Chemotherapy within the last 4 weeks (N = 855)	yes versus no	1.00	0.97 [0.41, 2.29]
Ongoing hormone therapy (N = 855)	yes versus no	0.43	1.30 [0.67, 2.53]
Ongoing mistetoe therapy (N = 855)	yes versus no	0.64	0.82 [0.36, 1.89]
Mistletoe therapy on the previous evening (N = 855)	yes versus no	0.69	0.89 [0.49, 1.59]
Hb (N = 812)	continuous	0.44	0.99 [0.98, 1.01]
Leukocytes (N = 813)	continuous	0.30	0.98 [0.93, 1.02]
Creatinine (N = 785)	continuous	0.01	0.31 [0.13, 0.78]
Gamma GT (N = 798)	continuous	0.26	1.00 [1.00, 1.01]
Concomitant medication (N = 855)	yes versus no	0.09	0.53 [0.25, 1.11]
Betablocker (N = 855)	yes versus no	0.65	1.25 [0.47, 3.29]
Thyroid hormones (N = 854)	yes versus no	0.07	0.53 [0.27, 1.04]
NSAID (N = 854)	yes versus no	0.33	0.72 [0.37, 1.40]
Metamizole (N = 854)	yes versus no	0.79	1.14 [0.45, 2.87]
Rosemary D3 (N = 855)	yes versus no	0.30	0.62 [0.25, 1.53]
Phosphorus D6 (N = 855)	yes versus no	0.01	0.506 [0.30, 0.86]
Secale cornutum/galena (N = 854)	yes versus no	<0.001	0.27 [0.14, 0.53]
Cardiodorone (N = 855)	yes versus no	0.59	1.25 [0.56, 2.80]
Bryophyllum (N = 853)	yes versus no	0.07	2.34 [0.95, 5.76]
Gelsenium (N = 851)	yes versus no	0.97	0.98 [0.48, 2.02]
Others (N = 853)	yes versus no	0.47	1.35 [0.60, 3.01]

Value effects are specified as odds ratios (ORs) with 95% confidence intervals. An OR = 1 indicates no correlation between the variables, while an OR > 1 indicates a higher probability of reaching the target temperature and an OR < 1 indicates a correspondingly lower probability of reaching the target temperature.

**Table 4 cancers-17-02716-t004:** Multivariate logistic regression (final model).

Parameter	Characteristics	*p*-Value	OR Estimator, 95% CI
Creatinine (N = 780)	continuous	0.01	0.30 [0.11, 0.78]
Secale cornutum/galena (N = 780)	yes versus no	<0.001	0.26 [0.12, 0.54]

Value effects are specified as odds ratios (ORs) with 95% confidence intervals. An OR = 1 indicates no correlation between the variables, while an OR > 1 indicates a higher probability of reaching the target temperature and an OR < 1 indicates a correspondingly lower probability of reaching the target temperature.

## Data Availability

Data will be made available from the corresponding author upon reasonable request.

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
