# Peer review of "Predictors of Successful Whole-Body Hyperthermia in Cancer Patients: Target Temperature Achievement and Safety Analysis"

_cancers, 2025, doi:10.3390/cancers17162716_

Round 1

Reviewer 1 Report

Comments and Suggestions for Authors

Dear editor

The manuscript entitled" Predictors of Successful Whole-Body Hyperthermia in Cancer Patients: Target Temperature Achievement and Safety Analysis" discusses about evaluation of factors for prediction of WBH treatment. This manuscript can be considered for publication after major revision and addressing following comments point-by-point.

1-The authors should add a graphical abstract or schematic figure which shows protocol of study. The manuscript is poorly provided in term of figures. 

2- The results are poorly written. It more contains tables with less explanations. It should be more improved and add more explanation.

3- What is novelty of your study? Explain exactly in introduction and compare with literature 

4- Conclusion should be improved.

Comments on the Quality of English Language

English should be polished.

Author Response

1-The authors should add a graphical abstract or schematic figure which shows protocol of study. The manuscript is poorly provided in term of figures. 

  • We thank the reviewer for this suggestion. We have now included a new schematic overview of the study protocol as Figure 1. This figure visually presents the study design, including patient inclusion, the timeline and procedure of moderate whole-body hyperthermia (WBH) sessions, and the subsequent outcome assessment steps.

Figure 1. Graphical summary of whole-body hyperthermia (WBH) outcomes in 397 cancer patients, showing high rates of target temperature achievement, low incidence of adverse events, and identification of serum creatinine and supportive remedies as key predictors of success.

2- The results are poorly written. It more contains tables with less explanations. It should be more improved and add more explanation.

  • We appreciate the reviewer’s observation. In the revised manuscript, the Results section has been substantially expanded to provide a clearer narrative alongside all tables. We now describe the key patient and treatment characteristics, summarize baseline values, explain the target temperature achievement rates, and highlight significant findings from both univariate and multivariate analyses. The adverse event profile is now presented with a textual overview emphasizing frequency, severity, and clinical context. We have also explicitly stated that all WBH treatments were performed outside of concurrent chemo- and radiotherapy phases to minimize confounding.

3- What is novelty of your study? Explain exactly in introduction and compare with literature 

  • We thank the reviewer for highlighting the importance of novelty and comparison with prior research. We have now clarified in the introduction that the present study is, to our knowledge, the first to systematically identify independent predictors of successful whole-body hyperthermia target temperature achievement in a consecutive real-world cohort, with a detailed safety profile analysis outside concurrent chemotherapy/radiotherapy. Previous studies predominantly addressed feasibility, combination therapies, or immunological parameters, but lacked comprehensive analyses of predictors and safety in standardized WBH protocols. We have updated the introduction and provided direct comparisons with major clinical reviews and trials.

Introduction ll. 90-107: Systematic reviews and phase II studies have described potential benefits of WBH as an adjunct treatment in advanced malignancies, but highlight significant limitations such as heterogeneity, small sample sizes, a lack of phase III data, and frequent use of combined therapy regimens [1,19–21]. Importantly, there is currently no consensus regarding in-dependent predictors of efficacy. For the first time, our study provides evidence-based insights into laboratory markers (e.g., serum creatinine) and pharmacological factors that predict successful achievement of target temperature, representing a substantive con-tribution to the optimization of WBH protocols and patient selection.

This study aimed to assess predictive factors for successful WBH treatment—defined as achieving a target core body temperature of ≥ 38.5 °C—and to describe the associated adverse effects. We retrospectively included all patients who underwent moderate WBH at a single integrative oncology center over a 12-month period. To our knowledge, this represents one of the first larger single center analyses to examine factors linked to target temperature achievement during moderate WBH in cancer patients, together with a structured safety assessment. By focusing on a consecutive real-world cohort treated in accordance with the German WBH guideline and outside concurrent chemo- or radiotherapy phases, our study provides practical insights into WBH application and tolerability in routine clinical practice.

4- Conclusion should be improved.

  • We thank the reviewer for this helpful suggestion. In the revised manuscript, we have rewritten the Conclusion to provide a more structured summary of our main findings, including the identification of serum creatinine and the use of secale cornutum/galena as independent predictors of achieving the target WBH temperature, as well as a concise description of the safety profile. We now also highlight the clinical relevance of these findings for patient selection and protocol optimization, place them in the context of current evidence, and note the need for future prospective research.

Conclusion, ll. 531-579: In this large single-center real-world analysis of 397 cancer patients undergoing a total of 855 moderate whole-body hyperthermia (WBH) sessions according to the German WBH guideline, we identified serum creatinine and the use of secale cornutum/galena as independent predictors of achieving the therapeutic target temperature of ≥ 38.5 °C. All treatments were performed outside of active chemotherapy and radiotherapy phases, allowing for an unconfounded assessment of safety. Adverse events were generally mild and transient, most frequently consisting of headaches, cardiovascular changes, and skin reactions.

Our findings add to the limited clinical evidence on factors influencing WBH effectiveness and may help guide patient selection and treatment planning in integrative oncology settings. In particular, the association between baseline creatinine and temperature achievement underscores the potential relevance of hydration status and renal function in optimizing WBH protocols.

While our retrospective design precludes causal inference, these results provide a basis for future prospective studies evaluating modifiable predictors of WBH success, refining patient preparation, and exploring the integration of WBH into multimodal cancer therapy.

Comments on the Quality of English Language

English should be polished.

  • The entire manuscript has been reviewed and the English improved throughout for clarity, conciseness, and flow.

Reviewer 2 Report

Comments and Suggestions for Authors

he author conducted a retrospective analysis and identified three key factors that may serve as predictors for the success rate and safety of whole-body hyperthermia (WBH). This study provides a potentially valuable reference for future WBH applications. However, the current content appears somewhat limited and does not yet provide sufficient depth to support publication as a full academic article. It is recommended that the author expand the manuscript to enrich its scientific value. In addition, I have several questions regarding this study and would appreciate it if the author could address them in the revised version.

  1. Certain chemotherapeutic agents can affect blood parameters, such as creatinine levels and leukocyte counts. How can the influence of these drugs on such data be minimized or avoided?
  2. Currently, approximately 50–60% of cancer patients receive radiotherapy during the course of their treatment. It is therefore recommended to include radiotherapy in Table 2 for further discussion.
  3. In this study, does the term "whole-body hyperthermia" refer to heating the entire body, or is the hyperthermia applied only to the tumor site?
  4. As indicated by the title "Predictors of Successful Whole-Body Hyperthermia in Cancer Patients: Target Temperature Achievement and Safety Analysis," the study identifies three factors that may serve as predictors for achieving the target temperature. However, it appears that no corresponding predictive factors were identified for the safety aspect.

Author Response

The author conducted a retrospective analysis and identified three key factors that may serve as predictors for the success rate and safety of whole-body hyperthermia (WBH). This study provides a potentially valuable reference for future WBH applications. However, the current content appears somewhat limited and does not yet provide sufficient depth to support publication as a full academic article. It is recommended that the author expand the manuscript to enrich its scientific value. In addition, I have several questions regarding this study and would appreciate it if the author could address them in the revised version.

1. Certain chemotherapeutic agents can affect blood parameters, such as creatinine levels and leukocyte counts. How can the influence of these drugs on such data be minimized or avoided?

  • We thank the reviewer for this valuable comment regarding the possible influence of chemotherapeutic agents on blood parameters such as creatinine and leukocyte counts. In our cohort, all WBH treatments were strictly performed outside of concurrent chemotherapy and radiotherapy phases in order to minimize the effects of these interventions on blood parameters and overall safety. Additionally, the variable “chemotherapy within the last 4 weeks” was included in our regression analyses to further adjust for any residual influences of recent cytotoxic therapy on laboratory values and outcomes. We have clarified these points in the revised manuscript.

ll. 170-172: All WBH sessions were scheduled during therapy-free intervals, i.e., outside of periods with ongoing chemotherapy or radiotherapy, to reduce confounding effects on laboratory and safety parameters.

ll. 202-204: All patients were treated as inpatients during chemotherapy-free and radiation therapy-free intervals, without receiving concurrent chemotherapy or radiation therapy during WBH sessions.

ll. 521-524: Furthermore, all WBH treatments were performed outside of concurrent chemo‑ or ra-diotherapy phases, ensuring that observed side effects were not influenced by additive acute toxicities from these treatments.

2. Currently, approximately 50–60% of cancer patients receive radiotherapy during the course of their treatment. It is therefore recommended to include radiotherapy in Table 2 for further discussion.

  • We thank the reviewer for this helpful suggestion. We have now included “radiotherapy within the last 4 weeks” in Table 2. In our cohort, 161 patients (40.6 %) had received radiotherapy at some time during the course of their disease. However, as specified in the revised Results section, all WBH sessions were conducted during chemotherapy-free and radiotherapy-free intervals, in order to avoid concurrent toxicity risks. We have also briefly commented on this in the Discussion section.

ll. 202-204: All patients were treated as inpatients during chemotherapy-free and radiation therapy-free intervals, without receiving concurrent chemotherapy or radiation therapy during WBH sessions.

ll. 521-524: Furthermore, all WBH treatments were performed outside of concurrent chemo‑ or ra-diotherapy phases, ensuring that observed side effects were not influenced by additive acute toxicities from these treatments.

3. In this study, does the term "whole-body hyperthermia" refer to heating the entire body, or is the hyperthermia applied only to the tumor site?

  • We thank the reviewer for this important question. We have now clarified in the manuscript that the WBH protocol applied in this study involves increasing the core temperature of the entire body, rather than localized tumor heating, in line with the Guideline on WBH of the German Society for Hyperthermia (DGHT).

ll. 68-71: Moderate WBH denotes therapeutic elevation of the body’s core temperature to thera-peutic levels through heating of the entire body (not localized tumor-specific hyper-thermia), typically between 38.5°C and 40°C, for a sustained period [4,5].

 ll. 156-158: This protocol involves heating of the entire body to achieve systemic elevation of core temperature, rather than localized hyperthermia focused on the tumor site.

4. As indicated by the title "Predictors of Successful Whole-Body Hyperthermia in Cancer Patients: Target Temperature Achievement and Safety Analysis," the study identifies three factors that may serve as predictors for achieving the target temperature. However, it appears that no corresponding predictive factors were identified for the safety aspect.

  • Thank you for this valuable comment. We agree that, unlike for the efficacy endpoint (target temperature achievement), no independent predictors for the safety aspect were identified. As adverse events in our cohort were generally mild and transient, no further exploratory analysis was conducted to identify predictors for toxicity, since this was not a primary aim of the study. We have now clarified this point in both the Results and Discussion sections.

ll. 310-314: Adverse events were generally mild to moderate in intensity, with most being transient headaches or minor cardiovascular changes. Given the mild overall toxicity profile, no additional exploratory analyses were performed to identify independent predictors for the occurrence of adverse events, as this was not considered a primary objective of the study.

  1. 524-529: While the study was designed to investigate predictors of successful target tem-perature achievement, it did not aim to systematically identify predictors for the occur-rence of adverse events. Given the overall mild toxicity profile observed, such an analysis was considered beyond the scope of the current work. Future studies involving larger cohorts and a higher incidence or severity of side effects may enable more meaningful modelling of potential toxicity predictors.

Round 2

Reviewer 1 Report

Comments and Suggestions for Authors

Manuscript is improved as well and can be published in present form.

Author Response

Thank you for your positive assessment and recommendation for publication. We appreciate the constructive comments that helped us improve the manuscript.

Reviewer 2 Report

Comments and Suggestions for Authors

In general, hyperthermia combined with radiotherapy or chemotherapy is administered within approximately two hours before or after the respective treatment. In this study, however, whole-body hyperthermia was applied only after the completion of the entire course of radiotherapy or chemotherapy. The rationale for adopting this treatment sequence remains unclear.

Author Response

Thank you for your valuable comment. In this study, whole-body hyperthermia (WBH) was intentionally administered only after the completion of the entire course of radiotherapy or chemotherapy, rather than concurrently or within a short time frame before or after these treatments. The rationale for this approach was to avoid additive acute toxicities and confounding effects from ongoing intensive therapies. By applying WBH outside of active chemo- or radiotherapy phases, the study aimed to ensure that observed side effects and temperature responses were attributable solely to WBH itself, thereby allowing for a clearer assessment of the safety profile and predictors of successful temperature achievement, without interference from acute treatment reactions. This sequencing provides more robust data on WBH feasibility and safety in an integrative oncology setting. 

To address this comment, we have revised the Introduction, Methods, Discussion, and Strengths and Limitations sections accordingly. These changes have been implemented in the manuscript and are highlighted in color for your convenience.